# Deep biogeographic barriers explain divergent global vertebrate communities

Peter J. Williams [1,2,3] ✉, Elise F. Zipkin [1,2] & Jedediah F. Brodie[3,4,5]

Biogeographic history can lead to variation in biodiversity across regions, but it remains unclear how the degree of biogeographic isolation among communities may lead to differences in biodiversity. Biogeographic analyses generally treat regions as discrete units, but species assemblages differ in how much biogeographic history they share, just as species differ in how much evolutionary history they share. Here, we use a continuous measure of biogeographic distance, phylobetadiversity, to analyze the influence of biogeographic isolation on the taxonomic and functional diversity of global mammal and bird assemblages. On average, biodiversity is better predicted by environment than by isolation, especially for birds. However, mammals in deeply isolated regions are strongly influenced by isolation; mammal assemblages in Australia and Madagascar, for example, are much less diverse than predicted by environment alone and contain unique combinations of functional traits compared to other regions. Neotropical bat assemblages are far more functionally diverse than Paleotropical assemblages, reflecting the different trajectories of bat communities that have developed in isolation over tens of millions of years. Our results elucidate how long-lasting biogeographic barriers can lead to divergent diversity patterns, against the backdrop of environmental determinism that predominantly structures diversity across most of the world.

The biotas of biogeographic regions contain unique species pools that have each evolved and assembled under different historical circumstances[1]. This means that regions with similar contemporary environments can differ substantially in species richness[2,3]. For example, glacial history explains differences in angiosperm diversity between East Asia and North America[4], past climate and plate tectonics explain why plant and vertebrate diversity is lower in the Afrotropics than in the Neotropics or Asian tropics[5], and differences in diversification and extinction rates across realms explain regional differences in brush-footed butterfly species richness[6]. Divergent evolutionary histories can also explain differences in functional diversity, including diets among desert small mammal communities[7] and functional composition among tropical savanna ant communities[8]. However, the legacy of biogeography on global biodiversity remains unclear. Although attempts have been made to elucidate the influences of particular historical effects such as diversification rates[9,10], past climatic conditions[5,11], and the movement of tectonic plates[12], there is a broader question to be addressed: to what extent do biogeographically distant communities differ in biodiversity metrics? That is, are communities that are more biogeographically isolated from one another more different in terms of taxonomic or functional diversity?

To study the effects of biogeography on biodiversity, biogeographic regions are typically treated as discrete, separate units, and analyses are either run individually for different regions[11,13–15] or include

[1]Ecology, Evolution, and Behavior Program, Michigan State University, East Lansing, MI 48824, USA. [2]Department of Integrative Biology, Michigan State University, East Lansing, MI 48824, USA. [3]Division of Biological Sciences, University of Montana, Missoula, MT 59812, USA. [4]Wildlife Biology Program, University of Montana, Missoula, MT 59812, USA. [5]Institute of Biodiversity and Environmental Conservation, Universiti Malaysia Sarawak, 94300 Kota Samarahan, Malaysia. ✉e-mail: peter.j.williams.110@gmail.com

'realm' or 'continent' as fixed or random effects[16–18]. However, regions vary greatly in how similar they are to one another. For example, in pan-tropical studies of animals, the Afrotropics, Asian tropics, and Neotropics are often treated as equally independent regions[5,11], even though the Afrotropics and Asian tropics share many more clades and evolutionary history with each other than either does with the Neotropics (Fig. 1). Just as taxonomy can be used to identify differences among species, realms can be used to identify differences among biotas. However, phylogeny is necessary to understand how evolutionary history has influenced present-day species. Likewise, to understand whether biogeographic history influences present-day diversity, we need to first understand the relationships among realms by quantifying the degree of shared biogeographic history, i.e., biogeographic distance.

Biogeographic distance among assemblages is best quantified using phylogenetic beta diversity turnover, or 'phylobetadiversity'[19]. Phylobetadiversity quantifies the evolutionary distance among assemblages using shared branch lengths, just as taxonomic beta diversity uses shared species[20]. The turnover component of phylobetadiversity partitions out beta diversity due to discrepancies in alpha diversity and only describes beta diversity due to 'replacement' or the presence of unique, unshared branch lengths after accounting for differences in phylogenetic alpha diversity[21]. This allows phylobetadiversity turnover to quantify shared evolutionary history at the assemblage level without being biased by species richness. Phylobetadiversity is increasingly used to define biogeographic regions through cluster analyses[19,22,23], including for terrestrial vertebrates[23], fishes[24], and trees[25]. However, we can also use phylobetadiversity turnover to represent continuous biogeographic distances among assemblages (Fig. 1). Biogeographic distances among local assemblages can then be used to evaluate differences in diversity among these assemblages. In contrast to using discrete realms to identify distinct regions that deviate from a global norm, the continuous approach using phylobetadiversity allows for the study of general patterns of biogeographic isolation on global or regional differences in biodiversity.

Here, we quantify the effect of biogeographic isolation on multiple facets of biodiversity in bird and mammal assemblages globally. We assess the relative strength of biogeographic isolation versus contemporary and past environmental factors in explaining global biodiversity patterns, specifically assemblage-level species richness, phylogenetic alpha diversity, and functional richness across 2°

latitude-longitude grid cells. While there is latitudinal variation in the area of our grid cells, other studies have found that grid cell area has little effect on inference in predicting species richness[26] or species distributions[27]. In addition to alpha diversity patterns, we further explore patterns of functional diversity by assessing the relationship between biogeographic isolation and mean pairwise functional beta diversity turnover to evaluate whether biogeographically isolated regions occupy distinct functional trait space, including diet, foraging stratum, and body size[28] (Supplementary Table 1). We choose these 'Eltonian' traits because they characterize the role and function of species within a community[28] rather than indirectly measuring function through morphological traits[29]. Ecological traits, such as those we used in this study, also tend to be less phylogenetically conserved than morphological or life-history traits[30], such that differences in evolutionary history among biogeographic regions might be expected to be less important for these traits.

## Results

### Bird and mammal alpha diversity

At the global scale, we found that alpha diversity tends to be deterministically related to contemporary environmental conditions, with biogeographic isolation playing a relatively small role, especially for birds. Human impact, the history of land conversion, the dissimilarity between past and present climates, and the history of ice cover each explained less than 1% of additional variance beyond a present-climate-only model (Supplementary Table 2). These variables were therefore excluded from final analyses. Biogeographic isolation alone explained 2.5–5.7% of the variance in species richness, phylogenetic alpha diversity, and functional richness, while environmental variables consisting of climate, elevation, topography, and landmass area explained 22.9–33.7% of the variance (Fig. 2). Most of the variance was shared between biogeographic isolation and environment, with 82.7–88.5% of the total variance explained in models that included both isolation and environment (Fig. 2). The variance explained by biogeographic isolation was consistently higher for mammals than for birds (Fig. 2). The strong influence of environment and the relatively weak influence of biogeographic isolation suggest that, on average, communities may have similar environmentally determined equilibrium diversity levels across regions[31,32], though this idea has been debated[33].

In contrast to these globally averaged findings, however, biogeographic isolation did strongly predict diversity in certain geographic areas. Biogeographic isolation greatly improved model fit for

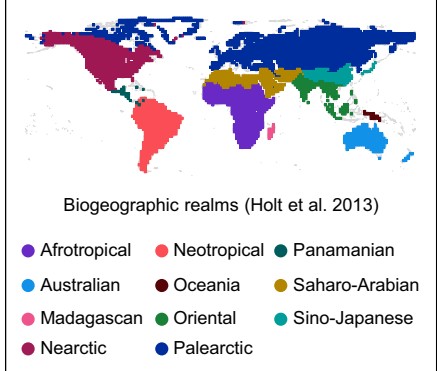

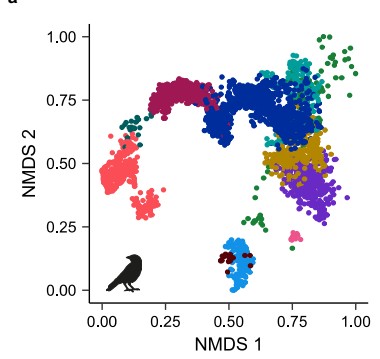

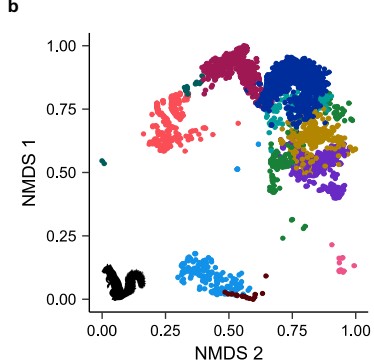

**Fig. 1 | Biogeographic isolation as quantified by phylobetadiversity.** Phylogenetic beta diversity turnover (phylobetadiversity) is commonly used to define biogeographic regions. Cells with low phylobetadiversity between them, represented by points plotted near each other in NMDS space, share more evolutionary history at the assemblage level and are biogeographically similar. Cells with high phylobetadiversity between them, represented by points plotted far from each other in NMDS space, are more biogeographically isolated. To illustrate how phylobetadiversity represents biogeography in a continuous manner, phy-lobetadiversity is shown for (**a**) birds and (**b**) mammals with points colored by biogeographic realm, as categorized by Holt et al.[23]. NMDS 1 and 2 of (**b**) are shown on the *y* and *x* axes, respectively, to aid comparison with (**a**) and with the map of realms. NMDS plots here are represented in two dimensions for the purpose of visualization (stress values 0.195 for mammals, 0.187 for birds), but three dimensions were used in our analyses (stress values 0.156 for mammals, 0.140 for birds). See Supplementary Fig. 4 for NMDS plots with three axes.

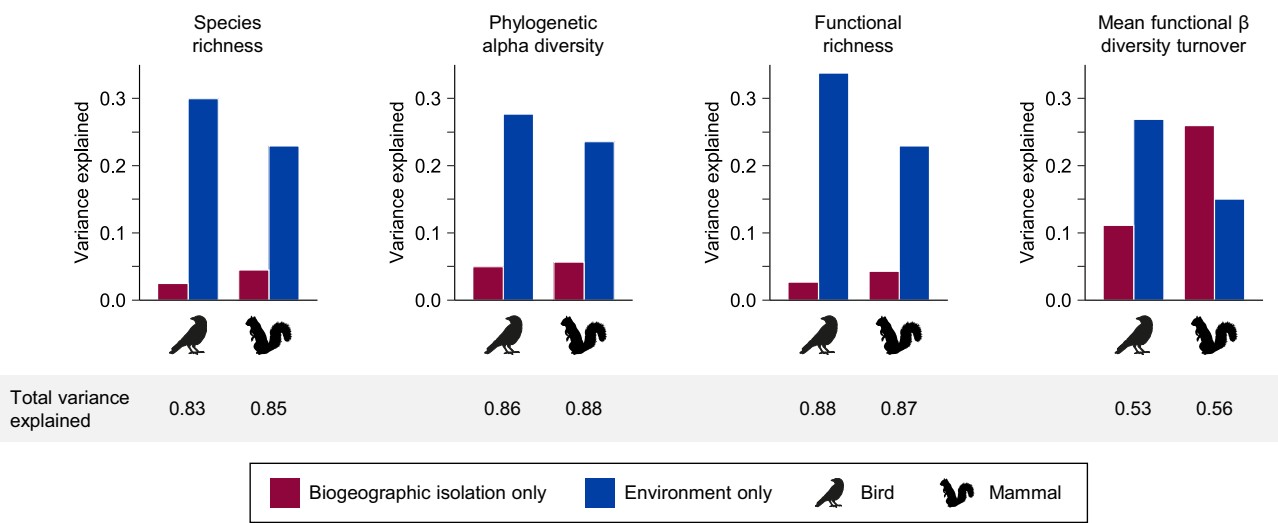

**Fig. 2 | On average, environmental conditions explain more of the variance in bird and mammal diversity than biogeographic isolation.** For each metric, total variance explained includes the variance explained by biogeographic isolation only (red), environment only (including climate, elevation, topography, and landmass area variables; blue), and shared between biogeographic isolation and environment (not shown). Variance explained was calculated by comparing adjusted $R^2$ values among a biogeographic isolation model, an environment model, and a global model that included both biogeographic isolation and environment variables.

mammal species richness in Australia and, to a lesser extent, New Guinea and Madagascar (Fig. 3a), as well as for mammal phylogenetic alpha diversity and functional richness in Australia (Supplementary Fig. 1). When using a discrete realm approach in place of biogeographic isolation, 'realm' also improved model fit for Australia, New Guinea, and Madagascar, though the effect of realm was much higher than the effect of isolation for New Guinea and Madagascar (Supplementary Fig. 2). That is to say, if New Guinea and Australia are both treated as independent units, then 'realm' is very important for New Guinea. Using a continuous approach to biogeographic isolation, though, acknowledges that Australia and New Guinea share substantial biogeographic history and are biogeographically quite similar. Therefore, additional factors besides isolation must explain New Guinea's atypical mammal biodiversity, in particular the low diversity of western New Guinea.

Australia, New Guinea, and Madagascar contain the most biogeographically isolated mammal assemblages in the world (Fig. 1b), and their deep isolation explains biodiversity patterns much better than does the environment alone. For example, the environment model predicted a median of 65.6 mammal species (range: 27.6–135.8) per grid cell on the Australian mainland, but the true median species richness is 35 (range: 17–101; Supplementary Fig. 3). The exact reason why Australian mammal richness is so low compared to other environmentally similar regions around the world is unclear. Australia has lost 20% of its terrestrial mammal species over the past 100,000 years, a much higher percentage than anywhere else in the world[34]. However, median species richness is 47% lower than predicted based on its environmental conditions, so recent extinctions do not fully explain Australia's low mammal diversity. In general, evolutionary time positively correlates with species richness[15], but Australian mammal assemblages are species-poor despite being evolutionarily very old[10]. Australian mammal species also have very low diversification rates on average, as calculated based on extant species, but diversification rates are often poor predictors of assemblage-level species richness[9,10]. Australia's low mammal diversity is therefore almost certainly due to its biogeographic isolation from other continents. Indeed, a repeated pattern in island biogeography is that isolation leads to low immigration of new clades, which can result in lower equilibrium species richness[35]. In addition to oceanic barriers, exchange between Australia and Southeast Asia has also been limited by differences in climatic tolerances of vertebrates that evolved on either side[36]. Apart from

Australia (and Antarctica, which has extremely low vertebrate diversity), all other continents are connected to another continent, allowing species to move among regions, while Australia has received far fewer new clades.

For birds, biogeographic isolation explained less variance for all biodiversity metrics than for mammals (Fig. 2). Biogeographic isolation did partially explain the low species richness of birds in Madagascar, but the effect of biogeographic isolation was inconsistent within other regions that are biogeographically isolated for birds, such as Australia and South America (Supplementary Fig. 1). Birds are much better than mammals at crossing oceanic barriers and many bird species cross oceans every year during migration. Nearly all of the largest bird orders are present in every realm, in contrast to mammals (Supplementary Table 3). However, there are still important examples in which biogeography influences the structure of bird communities. For example, woodpeckers and close relatives, the second largest taxonomic order in our dataset, are absent from Madagascar, Australia, and New Guinea (Supplementary Table 3). Suboscines (suborder Tyranni within Passeriformes) constitute over 10% of all bird species and yet the large majority of these species are Neotropical[37]. In contrast, songbirds (suborder Passeri within Passeriformes) originated in Australia but dispersed out of Australia multiple times[38], to the point where they are abundant worldwide, constituting almost half of all bird species, and many songbird clades are widespread. Even remote regions are less biogeographically isolated for birds than for mammals (Fig. 1), such that this comparatively limited isolation is not enough to substantially affect bird diversity.

## Functional beta diversity turnover

In addition to functional richness, which measures the total volume of trait space occupied, we evaluated how the distribution of functional traits differed among assemblages, i.e., whether biogeographically isolated regions contain unique sets of functional traits. We calculated pairwise functional beta diversity turnover, which is the degree of overlap between any two given assemblages in functional space and where the turnover component quantifies differences among assemblages that are not due to differences in alpha diversity (functional richness). For each assemblage, we calculated the mean value between the focal assemblage and all other assemblages. In other words, assemblages with high mean functional beta diversity turnover are functionally distinctive in that they occupy different areas of

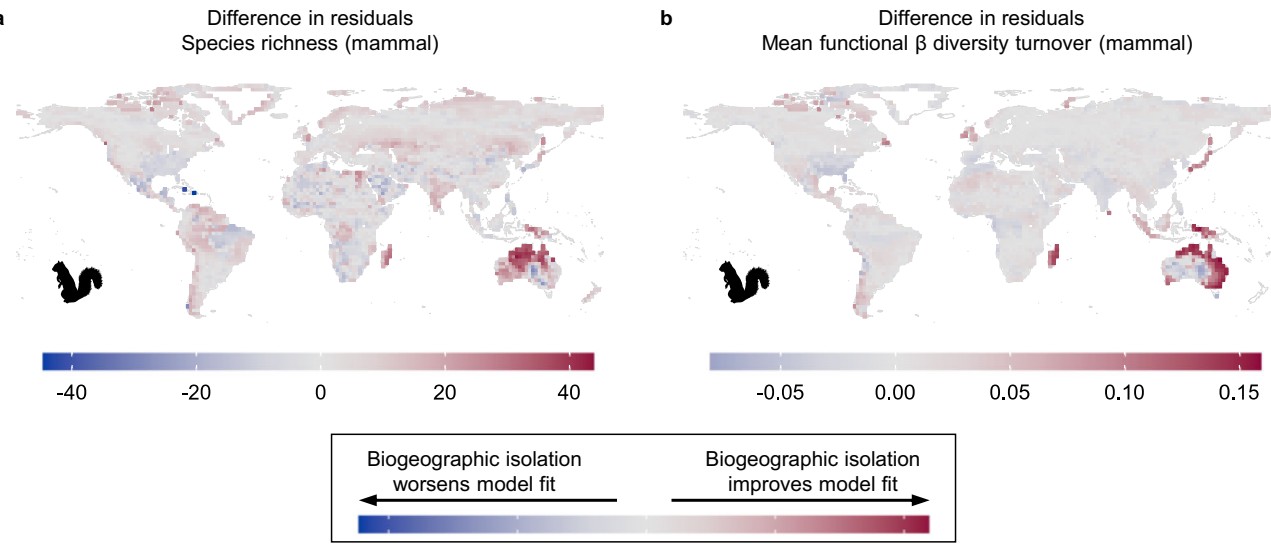

**Fig. 3 | Biogeographic isolation improves model fit for mammals in deeply isolated regions.** Differences in residuals were calculated as the absolute value of residuals in the environment-only model minus the absolute value of residuals in the global model using the residual values of each grid cell, shown here for (**a**) mammal species richness and (**b**) mammal mean functional beta diversity turnover. Scales differ for plots based on the units of the response variable. Positive values indicate that including biogeographic isolation improved model fit for grid cells, while negative values indicate that including biogeographic isolation worsened model fit. Maps for mammal phylogenetic alpha diversity and functional richness are very similar to panel **a** (see Supplementary Fig. 1). Maps are shown with an equirectangular projection.

functional space than most other assemblages. Overall, mean functional beta diversity turnover is quite low, especially for birds (Fig. 4), indicating that assemblages contain the same combinations of functional traits or are nested within the functional space of other assemblages. Unlike for birds, mean functional beta diversity turnover of mammals was best explained by biogeographic isolation (Fig. 2). The regions where mammal functional beta diversity turnover was highest are also many of the most biogeographically isolated regions: Australia, New Guinea, Madagascar, and the Caribbean (Fig. 4). Biogeographic isolation greatly improved model fit for many of these regions, especially Madagascar and the non-desert parts of Australia (Fig. 3). Biogeographic isolation also improved model fit for many islands and archipelagos that are not biogeographically isolated, which have low functional turnover similar to nearby mainland regions despite differences in landmass area. Our results thus reveal that the most biogeographically isolated regions of the world are also the most functionally distinctive.

There are several interesting examples illustrating how isolated mammal assemblages are functionally unique. In Madagascar, mammal assemblages contain arboreal frugivores, which are found only in tropical regions, but lack large ground-foraging herbivores, which are found almost everywhere around the globe including tropical, temperate, and even polar regions. Therefore, Madagascan assemblages occupy different functional space than other assemblages with similar environments or functional richness. These unique trait combinations are due to biogeographic isolation. With the exception of the now extinct Malagasy hippos (*Hippopotamus* spp.), no ungulates ever colonized Madagascar due to biogeographic barriers[39], leaving unoccupied functional space. Next, consider a pair of mammal assemblages, one from New South Wales, Australia, and one from the Eastern Cape, South Africa (Fig. 5). These assemblages have very similar climates, yet the Australian assemblage has, as reported, a much lower functional richness due to the lack of large carnivores and very large herbivores. Australia once had such megafaunal carnivores and herbivores, but they are now extinct[34]. However, the Australian assemblage also lacks scansorial granivores, though not due to recent extinctions. Murine rodents established in Australia around 4 million years ago[34], but sciurid and glirid rodents never did, leaving unfilled

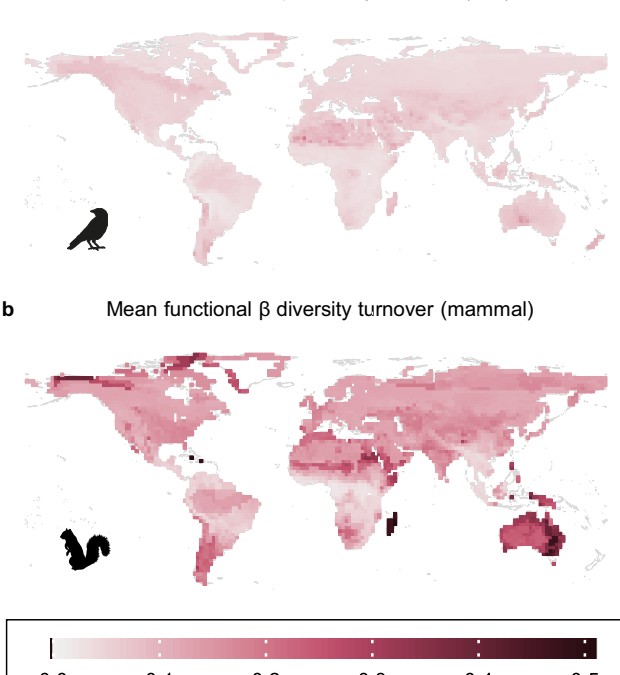

**Fig. 4 | Low bird functional beta diversity turnover and high variation in mammal functional beta diversity turnover across the globe.** Mean dissimilarity values of each cell for functional beta diversity turnover for (**a**) birds and (**b**) mammals. Functional beta diversity quantifies the pairwise functional dissimilarity between assemblages, which is partitioned into nestedness (differences in functional richness) and turnover (replacement of functional space in one assemblage by different functional space in another assemblage). Turnover values, calculated for each pair of cells, range from 0 (functional space of one assemblage is entirely nested within the functional space of another, or functional spaces of both assemblages are identical) to 1 (both assemblages occupy completely different functional spaces). Mean functional turnover of a cell is the mean functional turnover between one cell and all other cells, where high mean turnover indicates that a cell is functionally distinctive compared to other cells. Maps are shown with an equirectangular projection.

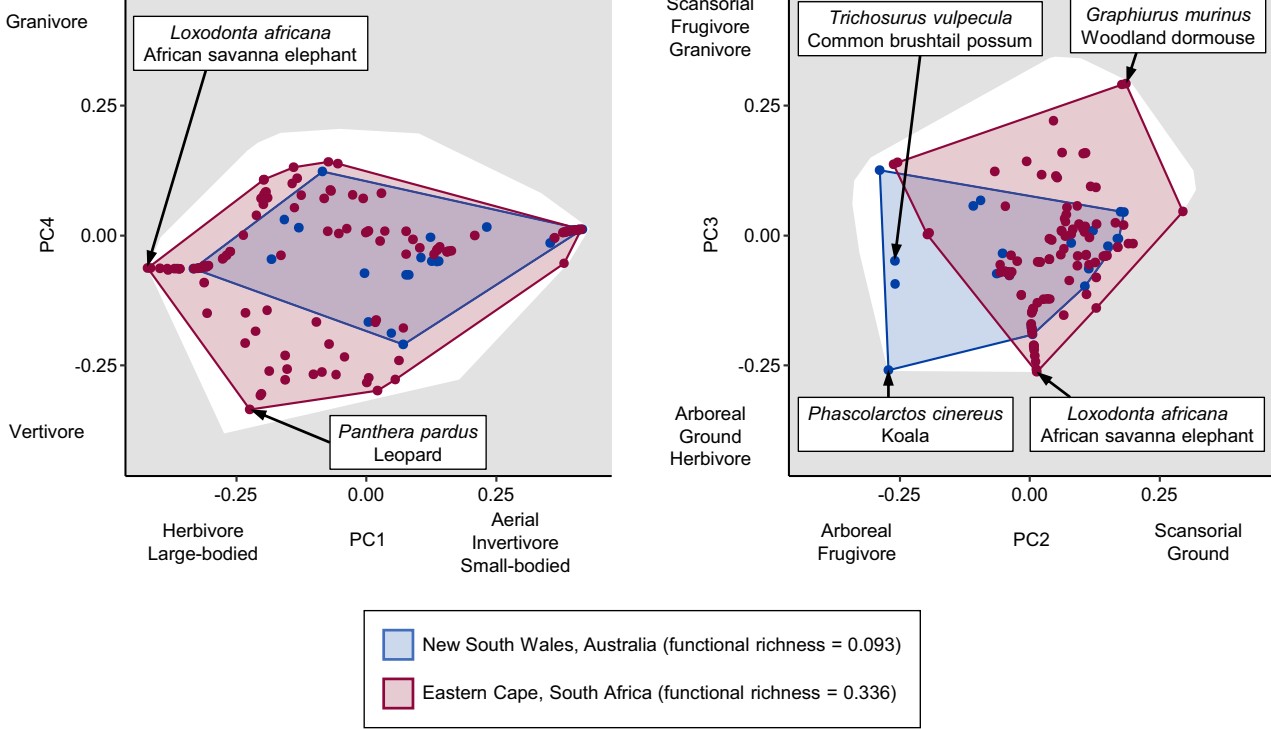

**Fig. 5 | High functional turnover between an Australian mammal assemblage and a climatically similar South African mammal assemblage.** The four PCoA axes shown were calculated using functional trait data from all mammal species. Functional richness is the proportion of total functional space occupied by an assemblage, where the white polygons represent the total functional space of all mammal species. Functional richness is the proportion of total functional space occupied by the convex hull of an assemblage. Functional beta diversity is the dissimilarity between two assemblages as measured by the overlap of the convex hulls of both assemblages. This example shows two assemblages from locations with very similar climates: the Australian assemblage has lower functional richness, but it occupies functional space that the South African assemblage does not, so there is relatively high functional turnover between the two assemblages.

functional space. In contrast, Australia does have koalas (*Phascolarctos cinereus*) and possums (Phalangeridae), while South Africa lacks such arboreal folivores. Unlike with birds, for whom biogeographic barriers between Australia and Asia have been weaker, the isolation of Australian mammals has led to functionally unique assemblages. The low functional richness and unique functional composition of Australia's mammals can help explain the high prevalence and devastating consequences of invasive mammals, which may have established by taking advantage of available functional trait space left unoccupied by native species[40].

## Bat diversity

To further examine whether the differences we detected between birds and mammals could be due to differential abilities to cross biogeographic barriers, we ran a separate analysis on the subset of mammals that are volant: bats. Through flight, bats have spread to many remote regions such as oceanic islands, like birds and unlike other mammals. This means that regions such as Australia are much less biogeographically isolated for bats than for other mammals (Supplementary Fig. 4). As predicted, and in contrast to mammals as a whole, biogeographic isolation did not improve model fit in Australia or Madagascar for bat alpha diversity or mean functional beta diversity turnover (Supplementary Figs. 1 and 5). When using a discrete realm approach, realm predicted bat diversity of New Guinea (Supplementary Fig. 2), though this effect is reduced in the continuous biogeographic isolation model that does not treat New Guinea and Australia as independent from each other. Interestingly, biogeographic isolation explained the striking difference in functional richness between Neotropical and Paleotropical bats (Fig. 6). The high functional richness of Neotropical bats is due to the incredible feeding diversity of

Phyllostomidae (leaf-nosed bats), a family endemic to the Neotropics that includes insectivores, vertebrate predators, frugivores, piscivores, and sanguinivores[41] (Supplementary Fig. 6). Having diversified in the Neotropics, members of Phyllostomidae were unable to disperse via higher-latitude routes to the Paleotropics due to an intolerance of cold[42], and the Afrotropics and Asian tropics still lack carnivorous, piscivorous, and sanguinivorous bats (Supplementary Fig. 6). This is one more example of how deep biogeographic barriers have led to differences in biodiversity patterns around the globe.

## Discussion

Overall, we find that biogeographic isolation does influence diversity patterns, but only when regions have been deeply isolated for a very long time. Australian mammals have been isolated from other large landmasses for 30–35 million years[43], which stands in contrast to other continents where land bridges currently or recently (e.g., Pleistocene) linked different regions. Tropical bat communities in the Eastern and Western hemispheres have been largely isolated from one another since the Eocene ~50 million years ago[41], but unlike other mammals, most bat clades could not cross the Beringia land bridge in the Pleistocene due to cold intolerance[42], so Neotropical and Paleotropical bats remain deeply isolated today. In contrast to mammals, many birds move between different biogeographic regions through annual migration, trans-oceanic dispersal, or an increased ability to traverse broad latitudinal or elevational gradients, reducing deep-time barriers. These results highlight the need to consider the biogeographic history of taxa to know whether there are deep-time barriers that may lead to differences in measures of biodiversity among regions.

In taxa or regions without deep, long-lasting barriers, we find that biogeographic effects are very weak, likely due to dispersal of clades

**a**

Functional richness (bat)

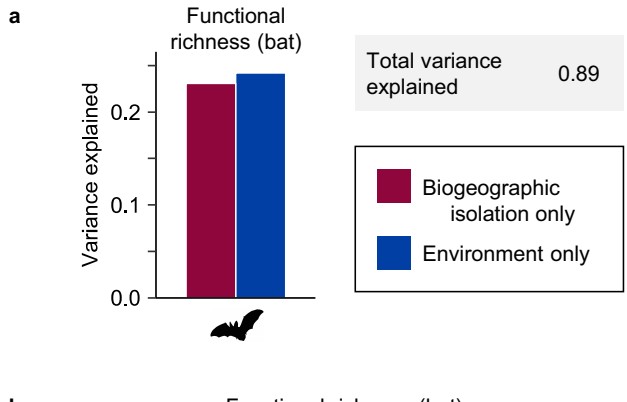

**b**

Functional richness (bat)

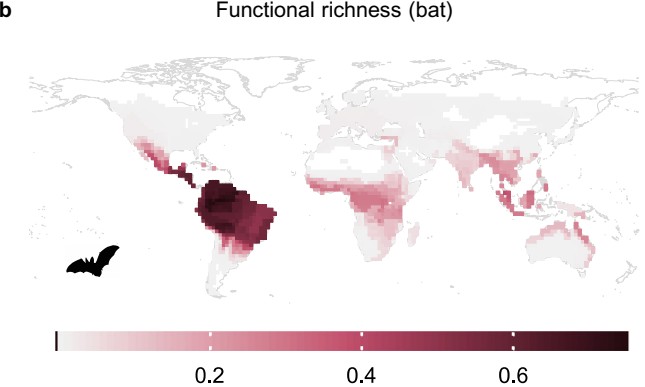

**Fig. 6 | Biogeographic isolation explains regional differences in bat functional richness between the Neotropics and Paleotropics. a** Variance partitioning results for bat functional richness. Total variance explained includes the variance explained by biogeographic isolation only (red), environment only (including climate, elevation, topography, and landmass area variables; blue), and shared between biogeographic isolation and environment (not shown). Variance explained was calculated by comparing adjusted $R^2$ values among a biogeographic isolation model, an environment model, and a global model that included both biogeographic isolation and environment variables. **b** Global pattern of bat functional richness. Map is shown with an equirectangular projection.

among regions. For birds and non-Australian mammals, beta diversity patterns at deep phylogenetic timescales are better explained by contemporary climate than by historic geographic isolation, meaning that dispersal has erased signatures of past isolation[44]. This may explain why biogeographic boundaries for birds are primarily determined by present-day climatic transitions, rather than by tectonic movements or other historical forces[45]. Apart from Australian mammals, most other bird and mammal lineages have dispersed and mixed to the extent that community composition is not limited by the availability of only a few lineages in any given area; this geographically widespread 'sharing' of lineages facilitates convergence among assemblages in different regions but with similar climates. For example, the functional similarities between temperate North and South American grassland mammal assemblages are likely due to the dispersal of North American clades into South America during the Great American Biotic Interchange[46,47]. Before the two continents joined, South America contained many evolutionarily unique clades, which may have resulted in communities that were functionally divergent from climatically similar North American communities[47]. Now, however, clades from North America have replaced many uniquely South American clades, and with the loss of isolation, the measures of biodiversity of South American mammals are in line with what we observe on other continents.

Our results highlight that deep biogeographic barriers may lead to differences among regions in species' diets and foraging strata traits. All of the traits we included showed significant phylogenetic signal,

though the strength of this signal varies and was, for most traits, lower than a null model of Brownian motion (Supplementary Table 1). Our results are consistent with those of a prior study of global mammal life history traits where phylobetadiversity turnover (what we call biogeographic isolation) predicted life history trait turnover[48], but our results dig deeper into the effects of isolation. The effect of phylobetadiversity in that prior study was not significant when compared to a null model that controlled for taxonomic turnover. However, when using phylobetadiversity as a representation of biogeographic isolation rather than trying to account for taxonomic differences across assemblages, there is a clear biogeographic signal for mammal life history traits[48]. This may be unsurprising given the reproductive differences between marsupials and placental mammals. However, it is intriguing that we found effects of biogeographic isolation using traits that are less directly tied to deep phylogenetic splits.

A central question of evolutionary biology has been how predictable evolutionary outcomes may be across species[49], and community ecology has been similarly plagued by questions of predictability[50]. Our results suggest that taxonomic and functional diversity around the world are broadly predicted by the environment. This may be facilitated by the dispersal of lineages, homogenizing regional taxonomic pools. However, if geographic areas have been isolated for long enough, such that regional species pools are very different, then biodiversity and community structure can vary greatly. If we replayed the tape of life, as Gould[51] put it, species composition across the world would undoubtedly differ greatly from the patterns observed today. But other than in deeply isolated regions, the mixing of species and clades in this counterfactual world would result in globally predictable patterns of biodiversity, resembling those observed today.

## Methods

### Species distribution data

Following Holt et al.[23], we used 2° × 2° latitude-longitude grid cells as the spatial units of our analyses, only including grid cells where land covered more than 50% of the cell. We excluded grid cells in Antarctica and the Greenland ice sheet that did not fall within one of Holt et al.'s realms. For each cell, we compiled lists of native extant and recently extinct mammal and bird species using range maps from the IUCN[52] and BirdLife[53]. We excluded marine mammals, pelagic birds, and portions of ranges where species were introduced or considered vagrant or transitory migrants. This resulted in sample sizes of 5008 and 9451 mammal and bird species, respectively, whose ranges overlapped at least one grid cell. All these species were included in further analyses. Taxonomies of the species distribution datasets were manually harmonized with the taxonomies of the trait datasets and the phylogenies. After accounting for taxonomic inconsistencies including changes to the species or genus name, lumping, and splitting, 1.0% of bird and 3.1% of mammal species were missing from the trait dataset, and 1.8% of bird and 0.2% of mammal species were missing from the phylogenies. The vast majority of these species were either extinct or recently discovered. For these species, we assigned trait values or phylogenetic position from the closest relative.

### Trait data

Functional trait data were taken from the EltonTraits database[28]. For mammals, the suite of functional traits included the percent of their diet consisting of invertebrates, fish, non-fish vertebrates, carrion, fruit, nectar, seeds, and other plant parts; foraging stratum (aerial, arboreal, ground, or scansorial; marine excluded); and body size (log-transformed). For birds, the suite of traits included the same diet percentage data as well as the percent of foraging strata consisting of aquatic, ground, understory, canopy, and aerial; and body size (log-transformed). Bats included the same traits as mammals, although carrion was not a part of the diet of any bat species.

We calculated phylogenetic signal of each functional trait using the *phylosignal* function in the *phytools* package version 1.9-16[54] and the *phylo.d* function in the *caper* package version 1.0.3[55]. For continuous traits, we calculated Blomberg's K[56], where 0 represents no phylogenetic signal and 1 represents Brownian motion. For categorical traits (mammal foraging strata) we calculated Fritz & Purvis' D statistic[57], where 1 represents no phylogenetic signal and 0 represents Brownian motion, but we rescaled our D statistic values to make them comparable with Blomberg's K.

### Environmental variables

We categorized the 'environment' of each grid cell using several variables. First, we used the Global Islands[58] dataset to determine the area of each island and continent, and we assigned a landmass area value for each cell. Second, we used GMTED2010 elevation data[59] to calculate both the mean elevation and elevational range or topographic relief of each cell. Third, we used WorldClim climate data[60] to calculate the mean values of each of 19 bioclimatic variables for each grid cell, and then we ran a principal component analysis (PCA) of all bioclimatic variables using the *prcomp* function in the *stats* package version 4.2.1[61]. Prior to running the PCA, we square root transformed precipitation variables to reduce skew. We used the first four principal components as our climate metrics, which cumulatively explained 93% of the total variation.

To test the potential influence of past climate, we created climate PCAs using WorldClim paleo climate data[60] for the Last Glacial Maximum and the Mid-Holocene. We ran analyses for bird and mammal species richness and functional richness using either present climate data, Last Glacial Maximum climate data, or Mid-Holocene climate data. We found that using paleo climate either lowered total $R^2$ values relative to present climate or increased $R^2$ very slightly (at most 0.8% additional variance explained; Supplementary Table 4). Therefore, we chose to use only present-day climate data for final analyses.

We tested several other variables that were not included in the final analyses. These variables included climatic distance between present and Last Glacial Maximum (measured as Euclidean distance in PCA space), climatic distance between present and Mid-Holocene, whether the cell was ever covered by an ice sheet[62], Human Impact Index[63], and years since significant land conversion[64] (as defined by Ellis et al.[65]). We ran analyses for bird and mammal species richness and functional richness and found that none of these variables increased $R^2$ >1% relative to a present day, climate-only model (Supplementary Table 2), so we did not include any of these variables in our final analyses.

### Phylobetadiversity

We used phylogenetic beta diversity turnover, 'phylobetadiversity', to characterize biogeographic distance between grid cells. Beta diversity turnover has been used to quantitatively define biogeographic regions[22,23], and phylobetadiversity turnover metrics such as $p\beta_{sim}$ have allowed biogeographers to rigorously describe biogeographic patterns[23,24,66], with $p\beta_{sim}$ the most commonly used metric[66]. We first created ultrametric consensus trees (including consensus branch lengths) based on 1000 credible phylogenies for all mammal[67] and bird[68] species. We calculated phylobetadiversity using the *phylo.beta.pair* function in the *betapart* package version 1.5.4[69] in R version 4.0.2[61]. Taxonomic beta diversity can be partitioned into two components: 'nestedness', the difference in diversity due to 'species loss' or discrepancies in species richness, and 'turnover', the difference in diversity due to 'species replacement' or the presence of unique, unshared species after accounting for differences in species richness[70]. Likewise, phylobetadiversity can be partitioned into nestedness and turnover components[21]. In the *phylo.beta.pair* function, the phylobetadiversity turnover component is measured using a phylogenetic extension of the Simpson's index or $p\beta_{sim}$[69], which uses phylogenetic branch lengths rather than species. We calculated $p\beta_{sim}$ for all grid cells, resulting in phylobetadiversity distance matrices for birds,

mammals, and bats. We then used the *metaMDS* function in the *vegan* package version 2.6.4[71] to perform non-metric multidimensional scaling (NMDS) on the $p\beta_{sim}$ distance matrices. For birds, mammals, and bats, we decided to use three NMDS axes based on stress values (stress = 0.156 for mammals, 0.140 for birds, 0.166 for bats; Supplementary Fig. 7).

### Response variables

We calculated assemblage-level species richness, phylogenetic alpha diversity, and functional richness for birds, mammals, and bats for each grid cell. We used the *pd* function in the *picante* package version 1.8.2[72] to calculate Faith's[73] phylogenetic diversity as the cumulative length of branches connecting the phylogeny of all species in a given grid cell. For functional richness, we first used the *mFD* package version 1.0.5[74] to calculate an n-dimensional PCoA for each taxon using functional trait data from the EltonTraits database[28]. We included four functional PCoA axes for birds and for mammals, but for bats three axes were sufficient. Using the *mFD* package, we calculated Villéger's[75] functional richness as the proportion of total functional space filled by the convex hull of an assemblage.

We used the *beta.fd.multidim* function in the *mFD* package version 1.0.1[74] in R version 4.0.2[61], to calculate pairwise functional beta diversity as the overlap of convex hulls in functional space using an extension of the Sørensen index of dissimilarity, which was then partitioned into a nestedness component and a turnover component. Nestedness here refers to non-overlap in functional space due to differences in functional richness. Since we already analysed variation in functional richness in this study, we disregarded nestedness and only focused on functional turnover—non-overlap in functional space due to assemblages occupying unique, unshared functional space that is not due to differences in functional richness. For each cell, we then calculated the mean functional beta diversity turnover between the focal cell and all other cells. Cells with higher mean functional beta diversity turnover occupy different functional space than other cells and are more functionally distinctive. We calculated functional beta diversity turnover separately for each taxon. To plot examples of overlap between assemblages in functional space, we used the *beta.multidim.plot* function in the *mFD* package[74].

A diagram of how variables used in analyses were derived from raw data products is provided in Supplementary Fig. 8.

### Linear regression and variance partitioning

To quantify the effect of biogeographic isolation on bird, mammal, and bat diversity, we used regression and variance partitioning. For each of our four response variables (species richness, phylogenetic alpha diversity, functional richness, and mean functional beta diversity turnover) and for each taxon (birds, all mammals, bats), we ran linear models where the predictor variables were climate PCA coordinates, mean elevation (plus a quadratic term), elevation range, landmass area (log-transformed), and phylobetadiversity NMDS coordinates. For climate, we included the first four PCA axes, with quadratic terms for each axis.

Unlike PCA axes, the position of a point along an NMDS axis is not meaningful; the distance between points in an NMDS reflects the dissimilarity of the points, and that distance is maintained even when axes are inverted or rotated[76]. Therefore, we treated phylobetadiversity as a three-dimensional surface and described biogeographic isolation using a second-order trend surface analysis. In practice, this meant our 'biogeographic isolation' variables included all three NMDS axes, quadratic terms for each axis, and interactions between each pair of axes. A trend surface describes a spatial pattern using X-Y coordinates (such as northing and easting)[76]. As with NMDS, the values of X and Y are not inherently meaningful, but they can be used to identify spatial gradients. Because trend surfaces describe spatial patterns, including a trend surface in a regression is one way to account for spatial

autocorrelation by modeling spatial structure[77,78]. Similarly, by using a trend surface for phylobetadiversity NMDS axes, we are modeling biogeographic structure. This method allows us to ask whether biogeographic isolation (distance within phylobetadiversity NMDS space) predicts metrics such as species richness. That is, do biogeographically close assemblages (points near each other in phylobetadiversity NMDS space) have similar diversity levels, and do biogeographically distant assemblages (points far from each other in phylobetadiversity NMDS space) have different diversity levels?

Biogeography is correlated with environmental variables such as climate, which is why we used variance partitioning to identify the variance explained by biogeographic isolation that is not explained by other variables. Besides the global model, we also ran models with only environmental variables (climate PCA axes, mean elevation, elevation range, and landmass area) or only biogeographic isolation (phylobetadiversity NMDS axes as described above). We then used variance partitioning to identify the variance explained by environment only, by biogeographic isolation only, and by both environment and biogeographic isolation. Linear models were run using the *lm* function in the *stats* package[61].

Finally, we assessed in which parts of the world biogeographic isolation improved model fit over an environment-only model by comparing the residuals of different models. We compared the residual values of each grid cell in the overall model that included both environmental variables and biogeographic isolation with the model that only included environment. If the residual of a grid cell in the overall model was closer to zero as compared to the environment-only model, then biogeographic isolation improved model fit for that grid cell. We calculated our 'difference in residuals' as the absolute value of residuals in the environment-only model minus the absolute value of residuals in the overall model.

To compare our continuous biogeographic isolation approach with a discrete realm approach, we reran all analyses replacing the set of phylobetadiversity NMDS axes with 'realm' from Holt et al.[23] as a categorical variable.

All data processing and analyses were performed in R version 4.2.1[61] except where noted.

## Reporting summary

Further information on research design is available in the Nature Portfolio Reporting Summary linked to this article.

## Data availability

Bird range map data may be requested from BirdLife[53] (http://datazone.birdlife.org/species/requestdis), and mammal range map data are available from the IUCN RedList[52] (https://www.iucnredlist.org/resources/spatial-data-download); we used range map data downloaded in 2018. Species trait data are available from EltonTraits[28] (https://doi.org/10.6084/m9.figshare.c.3306933.v1). Phylogenies for birds[68] and mammals[67] are available from VertLife (https://vertlife.org/data/). Climate data are available from WorldClim[60] (https://www.worldclim.org/data/worldclim21.html). Elevation data are available from the United States Geological Survey[59] (https://www.usgs.gov/coastal-changes-and-impacts/gmted2010). Landmass area data are available through the United States Geological Survey's Global Island Explorer[58] (https://rmgsc.cr.usgs.gov/gie/gie.shtml). Derived data products, including grid cell alpha diversity values and pairwise phylobetadiversity, are available on GitHub (https://github.com/pwilliams0/Biogeography_and_global_diversity) and Zenodo (https://doi.org/10.5281/zenodo.10779125).

## Code availability

All code for data processing and analysis are available on GitHub (https://github.com/pwilliams0/Biogeography_and_global_diversity) and Zenodo (https://doi.org/10.5281/zenodo.10779125).

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

## Acknowledgements

We thank E. Kopania for computational support and M. Newman for help in designing figures. We thank E. Kopania, J. Doser, S. Williams, and A. Mooers for critiques of previous drafts. This work was supported by the University of Montana, Michigan State University, and NSF grant DBI-1954406 (E.F.Z.). We thank F. Sayol, A. Wilson, and Y. Wong for the public domain animal silhouettes obtained courtesy of PhyloPic.

## Author contributions

P.J.W. and J.F.B. designed the study. P.J.W. performed analyses, analyzed results, and wrote the first draft of the manuscript. P.J.W., E.F.Z., and J.F.B. revised the manuscript.

## Competing interests

The authors declare no competing interests.
