## [Peer Review File · Nature Communications]

Deep biogeographic barriers explain divergent global vertebrate communitiesEditorial Note: This manuscript has been previously reviewed at another journal that is not operating a transparent peer review scheme. This document only contains reviewer comments and rebuttal letters for versions considered at Nature Communications. Mentions of the other journal have been redacted.

Reviewers' Comments:

Reviewer #2:

Remarks to the Author:

I have reviewed this paper twice already for REDACTED. My concern was mostly that it wasn't quite groundbreaking enough to be publishable in REDACTED, but I think it will be fine in Nat Comm. The few final comments I had have been satisfactorily addressed by the authors.

Reviewer #3:

Remarks to the Author:

I am happy with the revisions made for Nature Communications based on the first and second round of reviews at REDACTED. I have been enthusiastic about this study from the beginning. I note just a couple of minor things in this version:

line 61, 65 and elsewhere: the authors move back and forth between community and assemblage. I think the paper is about assemblages, not interacting communities, and though phylobeta metrics are agnostic as to the tips, this may cause confusion.

line 64: to the casual reader, the pendant edge leading to every unshared species in an assemblage (potentially contributing to a change in alpha diversity) would represent "unique, unshared branch length," and so this definition is, to my reading, incorrect.

I believe there is no simple way to describe the turnover component of these dissimilarity indices - the numerator is the PD unique to the less dispersed assemblage, but the denominator (PD in common - PD unique to the less-dispersed assemblage) is hard to put into words. I note the LaPriour et al paper that introduces phylobeta turnover has a lot of published corrections to the equations, which might give pause to readers who want to take a quick look. Given the concept and the metric is absolutely critical to the paper, how it is presented requires some more thinking.

Have folks ever made the comparison between Florida's herps and Australia's mammals before? I think I have read (by Pyron?) that one reason Florida has so many exotic herps is because its native fauna is depauperate due to its peninsular, semi-isolated shape.

We are grateful to the reviewers for reviewing our manuscript once more, and we are pleased that we have been able to satisfy their major concerns. We have responded to each of the reviewers' comments below in blue.

REVIEWERS' COMMENTS

Reviewer #2 (Remarks to the Author):

I have reviewed this paper twice already for REDACTED. My concern was mostly that it wasn't quite groundbreaking enough to be publishable in REDACTED, but I think it will be fine in Nat Comm. The few final comments I had have been satisfactorily addressed by the authors.

Response: We are glad that we have been able to address your concerns, and we appreciate you taking the time to review our manuscript for another round.

Reviewer #3 (Remarks to the Author):

I am happy with the revisions made for Nature Communications based on the first and second round of reviews at REDACTED. I have been enthusiastic about this study from the beginning. I note just a couple of minor things in this version:

Response: Thank you for your supportive and constructive reviews throughout the review process. We have addressed the specific points below.

line 61, 65 and elsewhere: the authors move back and forth between community and assemblage. I think the paper is about assemblages, not interacting communities, and though phylobeta metrics are agnostic as to the tips, this may cause confusion.

Response: We have replaced “community” with “assemblage” throughout the manuscript, including lines 65, 70, 74, 75, 206, 257, 259, 390, and 659.

line 64: to the casual reader, the pendant edge leading to every unshared species in an assemblage (potentially contributing to a change in alpha diversity) would represent “unique, unshared branch length,” and so this definition is, to my reading, incorrect.

I believe there is no simple way to describe the turnover component of these dissimilarity indices - the numerator is the PD unique to the less dispersed assemblage, but the denominator (PD in common - PD unique to the less-dispersed assemblage) is hard to put into words. I note the LaPriour et al paper that introduces phylobeta turnover has a lot of published corrections to the equations, which might give pause to readers who want to take a quick look. Given the concept and the metric is absolutely critical to the paper, how it is presented requires some more thinking.

Response: It is true that the turnover component is difficult to explain, even for taxonomic beta diversity, let alone phylogenetic beta diversity. We think the key is that turnover measures the proportion of unshared species/branches/etc. after accounting for differences in alpha diversity.

That is, two communities have species not found in the other, and these unique species are not just because one community has more total species than the other.

We have revised our definition of turnover in this section, as well as in the Methods, with the added text indicated by italics:

Line 66: “The turnover component of phylobetadiversity partitions out beta diversity due to discrepancies in alpha diversity and only describes beta diversity due to ‘replacement’ or the presence of unique, unshared branch lengths *after accounting for differences in phylogenetic alpha diversity*²¹.”

Line 360: “Taxonomic beta diversity can be partitioned into two components: ‘nestedness’, the difference in diversity due to ‘species loss’ or discrepancies in species richness, and ‘turnover’, the difference in diversity due to ‘species replacement’ or the presence of unique, unshared species *after accounting for differences in species richness*⁷⁰.”

Line 388: “Since we already analyze variation in functional richness in this study, we disregarded nestedness and only focused on functional turnover—non-overlap in functional space due to assemblages occupying unique, unshared functional space *that is not due to differences in functional richness*.”

Have folks ever made the comparison between Florida's herps and Australia's mammals before? I think I have read (by Pyron?) that one reason Florida has so many exotic herps is because its native fauna is depauperate due to its peninsular, semi-isolated shape.

Response: We are not aware of this comparison, but this is a fascinating idea that is worthy of a follow-up study!